# Biochemical Evaluation of *Agaricus* and *Pleurotus* Strains in Batch Cultures for Production Optimization of Valuable Metabolites

**DOI:** 10.3390/microorganisms10050964

**Published:** 2022-05-03

**Authors:** Dimitrios Argyropoulos, Charoula Psallida, Paraskevi Sitareniou, Emmanouil Flemetakis, Panagiota Diamantopoulou

**Affiliations:** 1Genetic Identification Laboratory, Institute of Technology of Agricultural Products (ITAP), Hellenic Agricultural Organization-Dimitra, 1 S. Venizelou Street, 14123 Lykovryssi, Greece; biomol_nagref@yahoo.gr (D.A.); chpsalida@yahoo.com (C.P.); vivian.sitar@gmail.com (P.S.); 2Laboratory of Edible Fungi, Institute of Technology of Agricultural Products (ITAP), Hellenic Agricultural Organization-Dimitra, 1 S. Venizelou Street, 14123 Lykovryssi, Greece; 3Laboratory of Molecular Biology, Department of Biotechnology, Agricultural University of Athens, 75 Iera Odos, 11855 Athens, Greece; mflem@aua.gr

**Keywords:** basidiomycetes, liquid cultures, glucans, glucanase, biomass, RNA

## Abstract

The production of various biochemical compounds such as proteins, glucans and glucanases, from the mycelium of four strains of Basidiomycetes species, *Agaricus bisporus*, *Agaricus subrufescens*, *Pleurotus eryngii* and *Pleurotus ostreatus*, during batch culture in shaking flasks, was studied. Fungi were cultured for 26 days in defined media with glucose as carbon source and were primarily evaluated for their ability to consume glucose and produce mycelial mass and intracellular polysaccharides (IPS). Results showed that on the 26th day of cultivation, *P. ostreatus* produced the maximum biomass (16.75 g/L), whereas *P. eryngii* showed the maximum IPS concentration (3.82 g/L). All strains presented a similar pattern in total protein production, with *A. bisporus* having the highest percentage of total proteins (36%, *w*/*w*). The calculated correlation coefficients among ribonucleic acid (RNA) vs. biomass (0.97) and RNA vs. protein (0.97) indicated a very strong relation between RNA and biomass/protein synthesis. The studied strains exhibited an increase in total glucan and glucanase (β-1,6) production during cultivation, with *A. bisporus* reaching the highest glucan percentage (8%, *w*/*w*) and glucanase activity (12.7 units/g biomass). Subsequently, processed analytical data were used in contour-graph analysis for data extrapolation to optimize future continuous culture.

## 1. Introduction

Mushroom cultivation is used for the synthesis of high-value biological products such as enzymes, polysaccharides and medicinal products with applications in environmental protection, novel antibodies, food and feed additives. A large number of different mushroom genera has been used for culture; however, the use of new mushroom species is an ongoing process as improved or new mushroom products are in constant demand, with *Agaricus* and *Pleurotus* being well-established in food and industrial uses [1].

The genus *Agaricus* includes more than 500 species, the most common being the edible *A. bisporus*, *A. campestris*, *A. arvensis*, *A. bitorquis* and *A. subrufescens* [2,3]. *Agaricus bisporus* has been cultivated for almost four centuries [4] and is found in nature in piles of manure and in living and dead plant tissues [5,6]. Its protein content varies from 24 to 44% of dry weight and its low fat content (1.7–8.0%, *w*/*w*) make it an ideal food with a positive effect on human health [7,8], combining high nutritional value with medicinal properties. *Agaricus subrufescens* was first cultivated in the late 18th century in Eastern North America; it is synonymous to *A. blazei* Murrill as it was found by genetic analysis [9]. The interest for the cultivation of this mushroom has increased significantly because of the presence of various bioactive molecules such as beta-glucans, with the main β- (1,6)–glucans, as also α- (1,6) and α- (1,4) [10,11]. Various application of β -glucans have been reported, e.g., their positive activity in hypercholesterolemic diet and their actions as immunomodulating agents with antitumor and anticancer properties and as regulators of gut microbiota [12,13,14]. 

The genus *Pleurotus* is one of the most widespread genera cultivated worldwide, including more than 200 species of mushrooms such as *P. ostreatus*, *P. pulmonarius*, *P. eryngii* and *P. sajor–caju* [15]. They have a high growth rate and utilize agricultural and industrial wastes as substrate for the production of high-nutritional- and pharmaceutical-value biomass, making them useful as commercial crops [16]. *Pleurotus* species are rich in proteins (10.5–30.4% d.w.) carbohydrates (57.6–81.8% d.w.), minerals (K, P, Na, Ca and Mg constitute the 56–70% of total ash content) and vitamins (e.g., thiamine and niacin are found at 1.16–4.8 and 46–108.7 mg/100 g d.w.) and are grown in a wide range of temperatures with an optimum between 18 and 30 °C [7,17]. The popular mushroom *P. ostreatus*, known as the oyster mushroom, contains β-(1,3)-(1,6)-glucans, which benefit human health [18,19]. *P. eryngii*, has excellent food characteristics and a variety of bioactive compounds (polysaccharides, peptides, fibers and lipids) with pharmaceutical properties [20].

However, in the referred mushrooms, nutritional compounds produced in solid substrates can be time-consuming and expensive and need large quantities of growth materials, large growth space and a specialized workforce [21]. For these reasons, the ability of mushrooms to produce a number of bioactive materials has caused the need to use liquid fermentation of mycelium for controlled and directed production of certain metabolites. Liquid fermentation is a biotechnological process, where all the physical and chemical agents required are provided in a well-defined form [22]. Usually, three main fermentation methods are applied in industrial applications, such as discontinuous/closed (batch), continuous and semi-continuous (fed-batch and cyclic fed-batch cultivation). These methods have many advantages over solid culture as they require less production time and space, the process can be automated and parameters such as temperature, humidity and pH are more easily controlled. The inoculation of the liquid medium can be achieved under strict aseptic conditions; mycelium grows more evenly and finally can be stored for a long time without genetic modifications [23,24]. The morphology of the mycelium, which is either pellets or filaments, depending on the method of cultivation (agitation rate, viscosity, etc.), and the genotype of the strain also affect the formation of biomass and specific products [25]. 

Popular carbon sources in liquid cultures are usually glucose and other simple sugars such as galactose, sucrose, maltose or mannitol [26,27,28]. Nitrogen sources used include organic sources such as amino acids, peptone, yeast extract and maltose, as well as inorganic sources (nitrate, nitrite and ammonia). For *A. bisporus*, glucose is proposed as the best carbon source, while for *A. subrufescens*, glucose, starch and the combination of glucose and dextrin [29,30]. As for the nitrogen source for both of these fungi, maximum mycelial growth was observed using yeast extract and peptone [31,32,33]. According to experiments performed by Gern et al. [34], in an attempt to find the proper composition of the growth medium of *P. ostreatus* to maximize the production of biomass and polysaccharides, the best results in terms of mycelial mass production were observed when glucose was used at a concentration of 40 g/L and corn extract at a concentration of 20 g/L as a source of organic nitrogen.

Liquid mushroom cultivation has been focused on carbohydrates and especially on polysaccharide production, which as functional foods have been considered as prebiotics [35]. These polysaccharides are divided into intracellular polysaccharides (IPS) (mainly glycogen, pullulans, β-glucans) and disaccharides such as trehalose and extracellular polysaccharides or exopolysaccharides (EPS). They play a special role in food, and the biotechnological and pharmaceutical industries with huge economic significance [36,37]. Fungal mycelium also has high protein content and is considered as a reliable vegan protein source. The proteins present a high protein-to-energy ratio, additional nutritional benefits and an undoubted potential to decrease meat consumption and the carbon footprint [38]. *Pleurotus* spp. are a typical example of liquid cultures whose biomass contains a high percentage of protein. With a mean range of protein content from 10.5 to 49% of dry weight, it is important to investigate the nutritional value of these proteins, as well as different ways to increase their production [39]. On the other hand, cultivation of *Agaricus* spp. presents a wide range of protein concentrations depending on medium composition [40,41]. Indicative of the economic value of mushroom protein content in submerged cultures is the effort to design industrial methods using *A. bisporus* mycelium for the production of a highly acceptable meat analogue, having, according to authors, not only superior textural properties but also umami characteristics when compared to that of soy protein [42]. Beyond such efforts, amino-acid analysis seems to be dependent to strains and different growth conditions, but understanding the factors influencing mycelia protein production it is not clear yet [24]. 

In addition to structural-compound production, mycelium fermentation is induced by a wide variety of liquid substrates to produce enzymes applicable in specific biotechnological bioprocesses [43]. Enzymes such as lignin peroxidase, manganese peroxidase, laccase and hydrolytic enzymes such as cellulases, hemicellulases and xylanases are secreted and isolated from the growth medium [44,45]. In recent studies, acidic thermophilic cellulase has been purified from the *P. ostreatus* submerged cultures with hydrolytic potential on selected agro-wastes, making it an ideal candidate for the bioconversion of biomass in second generation biofuel production [46]. Several additional studies highlight submerged cultivation of *Pleurotus* mushrooms for producing hydrolytic enzymes and achieving a high cellulase yield [43,47,48]. Similarly, many *Agaricus* species have also been reviewed for the production of glucanases [49].

Recently, with improvements in nucleic analysis, mycelium RNA has attracted attention for the study of RNA and especially of certain fractions that are part of the total RNA content [50]. While there is active research on specific RNA species in fungal regulatory mechanisms, total RNA relation has not presented extensively. More recent work for growth rate and biomass relating to RNA is found with yeast cultures and algae [51,52]. In this study, total RNA is correlated to protein and biomass production of *Pleurotus* and *Agaricus* as an indicator of fungal growth dynamics. 

Four strains of the popular mushroom genera *Pleurotus* and *Agaricus* were tested in batch cultures to determine behaviors in enzyme and polysaccharide production along with other biological constituents for potential industrial and commercial applications. Additionally, processed data were also supported by contour graphical analytics for future continuous culture optimization. The purpose of the present study was the evaluation of the referred strains in order to be used in future production of valuable metabolites, as well as to establish a framework for similar assessments of other promising mushrooms maintained in the collection of the higher fungi stored at the Athens Mushroom Research Laboratory (AMRL).

## 2. Materials and Methods

### 2.1. Biological Material, Media and Culture Conditions

In the current work, the edible basidiomycetes *A. bisporus* AMRL 204, *A. subrufescens* AMRL 235 (CA 560), *P. eryngii* AMRL 173-6 and *P. ostreatus* AMRL 137 were studied. The strains are stored at AMRL in the fungal culture collection of the Edible Fungi Laboratory (ELGO-Dimitra) and are maintained in potato dextrose agar (PDA, Merck, Darmstadt, Germany) at 2 ± 0.1 °C. 

Cultures were performed in 100 mL Erlenmeyer conical flasks, filled with 30 ± 1 mL growth medium. High-chemical-grade [D(+) glucose (Alfa Aesar, Kandel, Germany) was used as the main carbon source, while peptone (Merck, Darmstadt, Germany) and yeast extract (Fluka, Steinheim, Germany) were used as the main nitrogen source, as reported by Diamantopoulou et al. [36]. The analytical composition of the growth substrate, common in all cultures, is (g/L): Glucose 30.0; Yeast extract 2.5; Peptone 3.5; CaCO_3_ 0.1; KH_2_PO_4_ 1.0; MgSO_4_∙7H_2_O 0.5; CaCl_2_∙2H_2_O 0.3; MnSO_4_∙H_2_O 0.04 and ZnSO_4_∙7H_2_O 0.02. Yeast extract contained 11%, *w*/*w* nitrogen and 10%, *w*/*w* carbon, whereas the respective quantities in peptone were 11% and 35%, *w*/*w* and were taken into consideration for the estimation of the C/N ratio in the growth medium (C/N = 20). The conical flasks with the culture medium were sterilized in autoclave (121 ± 0.5 °C, 20 min) and inoculated with 7 mm agar plugs cut from a growing colony in a PDA Petri dish, aged 5–10 days, depending on the strain. The initial pH for all media before and after sterilization was 6.4 ± 0.1 °C. The initial dry biomass concentration was estimated to be 0.10 ± 0.02 g/L. The cultures were incubated at 26 ± 1 °C for 26 days on a rotary shaker (ZHICHENG ZHWY 221C, Shanghai, China), with a stirring rate of 120 ± 5 rpm. 

### 2.2. Biomass Collection and Determination

Biomass was collected after vacuum filtration, using filters (Whatman No 2, Buckinghamshire, UK) followed by triple rinsing of the biomass with deionized water [36]. The filtrate was stored in freezing conditions (−20 ± 1 °C), while the fresh mycelium was transferred to pre-weighed McCartney glass vials and the fresh weight of the mycelia was recorded with a four-digit balance (Kern AGB, Breisgau, Germany). This was followed by drying at 60 ± 1 °C for 2–3 days, after which the dry weight of the mycelium (biomass) was recorded again. For the biomass determination three replications were used for each sample. 

### 2.3. Determination of Reducing Sugar Consumption

Determination of nonconsumed reducing sugars by the fungi in the nutrient media took place by the photometric method of dinitrosalicylic acid [53]. Specifically, 0.5 mL of DNS reagent was added to 0.5 mL sample (diluted 1:25), followed by stirring in the Vortex apparatus. The samples were transferred in a water bath at 100 ± 1 °C for 5 min and then left at 25 ± 0.5 °C. Finally, 5 mL of deionized water was added, stirred in the Vortex apparatus and the absorbance measured at 540 nm on a dual-position UV-Vis spectrophotometer (Jasco, V-530 UV/Vis Spectrophotometer, Tokyo, Japan). The concentration of samples in reducing sugars was calculated from a standard reference curve, expressed in g/L glucose.

### 2.4. Determination of Intracellular Polysaccharides (IPS)

According to Diamantopoulou et al. [36], 100 mg of dry biomass was extracted using 20 mL of 2.5 M HCl (Merck, Darmstadt, Germany) for one hour at 100 ± 1 °C. Then, the pH of the extract was adjusted with 2.5 M NaOH (Merck, Darmstadt, Germany) and the final volume was adjusted with deionized water to 100 mL. Total sugars were quantified by the DNS method and IPS were expressed in glucose equivalents using a glucose-concentration standard curve. 

### 2.5. Determination of Proteins

For the determination of proteins, a spectrometric method was used with the help of Bradford reagent (Sigma, Steinheim, Germany), according to the Bradford method [54]. A standard curve was prepared using Bovine serum albumin (BSA) in serial dilutions for the determination of samples’ protein concentration. In 10 μL of each sample 240 μL of Bradford dye was added (three repetitions for each sample) and the corresponding absorbance was determined on ASUS expert 96 (Biochrom, Cambridge, UK) microplate analyzer at 620 nm.

### 2.6. Determination of RNA Levels

The RNA genetic material of the samples was extracted from 5 mL lyophilized samples using the RNA for plants and fungi kit (Nucleospin, Macherey-Nagel) and quantified by AccuBlue^®^ Broad Range RNA Quantitation Kit with fluorescence spectrometry according to manufacturer’s protocol. Crude RNA released from 1 mL lyophilized sample in 0.2 mL TES buffer with 1% CTAB after centrifugation (12,000 rpm, 10 min) was visualized by electrophoresis on a 2% *w*/*v* agarose gel in a horizontal electrophoresis apparatus using a Vilber Lourmat CCD camera [55].

### 2.7. Determination of Total Glucans

The determination of the total glucans in the samples (100 mg dry mass) was performed according to the protocol of the commercial diagnostic preparation kit: Mushroom and Yeast beta-glucan K-YBGL (Megazyme, Wicklow, Ireland) [56]. 

### 2.8. Determination of Glucanase Activity

Glucanase activity was determined using the commercial polysaccharide CM-Curdlan (Megazyme, Wicklow, Ireland) according to manufacturer’s protocol. The certain polymer is completely soluble to give an excellent substrate for the assay of endo-1,3-β-D-glucanase by reducing-sugar or viscometric procedures. For each sample case, 250 μL were obtained in triplicates and 250 μL of CM-curdlan solution (10 mg/mL) was added, followed by incubation for 20 min. Reducing-sugar measures corrected against blank samples and glucose equivalents were determined using DNS method [53] and a standard glucose curve. Activity was expressed as unit/g biomass.

### 2.9. Determination of Growth Rates

Growth rates (μ) of the studied fungi along the corresponding biomass growth curve were numerically calculated using dry weight and culture time through the formula:μ = [ln(biomass at t_2_) − ln(biomass at t_1_)]/(t_2_ − t_1_)
where t_1_ and t_2_ refer to sampling time points. Maximum growth rate (μ_max_) refers to the maximum μ value, determined at the exponential phase of biomass production [57].

### 2.10. Optimization Analysis

Extrapolated optimization analysis in silico was performed with Design Expert (ver. 13, 2021) [58]. Data obtained from the batch cultures set the basis to identify culture growth rates and carbon concentrations for optimum production of glucans and glucanases in future continuous fermentations.

### 2.11. Statistical Analysis

Differences were evaluated at a 95% confidence level (*p* ≤ 0.05) using one-way analysis of variance (ANOVA) followed by the pairwise *t*-test. Statistical analysis was performed using the Statgraphics Centurion XVII statistical software.

## 3. Results and Discussion

Four strains of the edible basidiomycetes *Agaricus* and *Pleurotus* were studied in batch cultures during a period of 26 days for quantification of biomass production, substrate consumption, (total) IPS, proteins, RNA, (total) glucans and endo-1,3-β-D-glucanases. Comparative results about their biochemical behavior are presented, in order to identify the suitable strains in relation to their enzyme and complex carbohydrate production. 

### 3.1. Biomass Production–Glucose Consumption of the Substrate

Based on previous data concerning the cultivation of several mushrooms in liquid-state cultivations and the positive effect of agitation on biomass production, the present cultures were grown on mild agitation speed [59]. Kinetic data of biomass production of *A. bisporus*, *A. subrufescens*, *P. eryngii* and *P. ostreatus* are presented in Table 1. It is observed that in all cases the maximum biomass production was attained during the last days of cultivation. *A. bisporus* and *P. ostreatus* reached the highest values on the 21st and 26th day, 13.06 and 16.75 g/L, respectively. This observed max value of *P. ostreatus* was significantly different from the corresponding values of the rest of the strains. Comparative evaluation between *Agaricus* strains showed that *A. bisporus* produced a much higher, statistically significant amount of biomass than *A. subrufescens*, 12.67 vs. 5.73 g/L, even at the early stages of fermentation, presenting a significant increase in biomass production between the 12th and the 16th day. On the other hand, its max biomass value was not significantly different when compared to that of *P. eryngii*. Towards the end of the cultivation period, *A. bisporus* reached the stationary phase of its growth, while *A. subrufescens* showed a low but still increasing production of biomass. Regarding the genus *Pleurotus*, both strains showed a similar growth profile, but *P. ostreatus* produced a higher biomass amount at all sampling points, reaching the maximum value (16.75 g/L) at 26th day. Since the medium and culture conditions were the same for all fungi, the differences observed in their biomass production are attributed to the genetic background of the fungi, as each strain has a different way of exploiting the substrate and a different rate of mycelial growth. The observed behaviors are also verified from the calculated maximum growth rates (μ_max_) for each of the four fungi, *A. bisporus*, *A. subrufescens*, *P. eryngii* and *P. ostreatus*, are 0.28/d, 0.14/d, 0.22/d and 0.29/d, respectively.

Glucose was used as the main carbon source during fungal growth for biomass and metabolite production. Residual glucose (g/L) results are also shown in Table 1. It is observed that strains *A. bisporus* and *P. ostreatus* consumed the greatest amount of glucose (96 and 99%, respectively) indicating that the fermentation was almost completed at the end of the cultivation time (26 days), while the slow-growing *A. subrufescens* and *P. eryngii* failed to utilize the provided glucose within the certain culture period (consumed only 46 and 57%, respectively). Results also show that there was potential for further continuation of the cultivation, as not all of them had entered the stationary phase of their growth (e.g., *P. eryngii* and *A. subrufescens*), but for reasons of comparative study and economy all fungal cultures were terminated simultaneously. *A. subrufescens* and *P. eryngii* showed rapid glucose consumption for the first 6 days and then their consumption ability decreased. Conversely, *A. bisporus* and *P. ostreatus* showed rapid glucose consumption in the entire culture period. It is evident that glucose consumption of the substrate differed between species, as it was totally consumed for two of the four fungi studied (*A. bisporus* and *P. ostreatus*), whereas *A. subrufescens* and *P. eryngii* consumed part of it. Biomass synthesis was accompanied by yield Y_Χ/Glc_ (yield of mycelial dry biomass produced per unit of glucose consumed) values that were high for *P. eryngii* and *P. ostreatus* (0.72 and 0.60 g/g respectively). Therefore, the rate of substrate utilization for biomass production was not genus but species dependent in the studied cases. The maximum-yield Y_X/Glc_ values achieved in the current research were significant, given that the reported maximum theoretical yield of dry biomass produced per gram sugar consumed is ~0.8 g/g [60]. However, in a previous study of Diamantopoulou et al. [59], *P. ostreatus* AMRL 150 cultivated at similar growth medium and conditions presented higher yield (Y_X/Glc_ = 0.83) probably because the cultivation lasted only 16 days. 

In similar experiments with the same fungal species, it was also noticed that there is great variation in the produced biomass. For example, Sarris et al. [61] reported very low values of biomass produced the 16th day of cultivation by *A. bisporus* AMRL 208 and 209 (2.8 and 3.3 g/L, respectively) in static flasks with 30 g/L glucose, accompanied by low glucose consumption. When Lima et al. [62] cultivated *A. subrufescens* in medium containing 20 g/L glucose and 4 g/L yeast extract, a biomass value of 7.8 g/L was observed. Similarly, Liu and Wang [63] used a mixture of glucose and corn extract as a source of glucose and wheat bran and yeast extract as a source of nitrogen in a 30 L bioreactor with an agitated rate of 160 rpm, and a biomass value of 11.30 g/L was then recorded. In experiments with five species of *Pleurotus* grown for 15 days in a medium containing 20 g/L glucose, 2 g/L peptone and 3 g/L yeast extract, *P. eryngii* produced 6.1 g/L of biomass, whereas *P. ostreatus* produced 11.2 g/L [64]. Kim et al. [27] cultured *P. ostreatus* in conditions similar to the present study and observed smaller values of produced biomass on the 5th, 10th and 15th culture days—1.07, 4.6 and 7.35 g/L, respectively. In addition, El-Enshasy et al. [65], cultured *P. ostreatus* in agitated flasks filled with 20 mL of the same nutrient as in the present study but at lower concentrations [glucose (20 g/L), peptone (0.5 g/L) and extract yeast (2 g/L)] and observed that the maximum biomass, 4.5 g/L, was produced on 9th day. In contrast, when the fungus was grown on a substrate containing glucose (40 g/L) and extract (20 g/L) in flasks at an agitation rate of 120 rpm, the weight of the dry biomass was 20.49 g/L [34,36]. Finally, Diamantopoulou et al. [59] achieved 19.2 g/L of biomass along with very high glucose consumption, when *P. ostreatus* AMRL 150 was grown for 16 days in a similar medium to this study. 

The reported in literature variation in biomass production indicates that beyond genetic differences among strains of the same genus, there is also a need for additional studies to optimize media composition and culture conditions in order to establish large-scale continuous cultures for potential commercial production. 

### 3.2. IPS Determination

Mean absolute (g/L) and relative (mg/g) values of IPS produced by strains *A. bisporus*, *A. subrufescens*, *P. eryngii* and *P. ostreatus* are presented in Table 1. Data showed that IPS (g/L) increased as a function of cultivation time for the fungi *P. eryngii* and *P. ostreatus*, reaching the maximum values of 3.82–3.56 g/L, whereas in *A. bisporus* cultures the respective values were stabilized between the 16th and 21st day and reduced afterwards, probably due to degradation. For *A. subrufescens*, a small amount of IPS was produced at the end of the cultivation period (1.3 g/L). It is also observed that all cultures resulted in IPS ranging between 20 and 40% *w*/*w* of the total biomass. Specifically, *P. eryngii* synthesized the highest IPS amount (37.97%, *w*/*w*) exhibiting a small decline in the last days of culture, followed by *A. subrufescens*, *A. bisporus* and *P. ostreatus*. However, as reported by Krüzselyi et al. [66], the carbohydrate content in mycelia was lower (55.36%, *w*/*w*) compared to that of 62.77% in pileus and 78.36% *w*/*w* in stipes of fruiting bodies of *P. eryngii*. Additionally, it was observed that although *A. subrufescens* produced the lowest biomass of all strains, it produced a high IPS amount (23–35%, *w*/*w*), similar to those of *A. bisporus* (20–32%, *w*/*w*). It is also noticeable that in all cases, the highest values of IPS were produced in the early stages of cultivation, when biomass concentration was low, and as the mycelium mass was increasing, the concentration of IPS was decreasing. This indicates that part of IPS was used for cell structures, such as cell walls, leading to the observed increase in biomass. IPS accumulation and degradation, occurring when growth was limited by glucose depletion has been also reported in liquid cultures on glucose-based media with C/N 20 for *Volvariella volvacea* [67,68], *Morchella esculenta* and *Flammulina velutipes* [36], *Tuber sinense* [69], *Ganoderma lucidum* [70] and *Grifola umbellata* [71]. Different bioactive compounds from mushrooms, and particularly polysaccharides, are resistant to the acidic environment of the stomach and the human digestive enzymes and thus are able to participate in various interactions with gut microbiota, indicating their potential application as prebiotics and dietary fiber [72]. 

### 3.3. Protein Production

The results of protein production of the fungi (in absolute - g/L and relative - %, *w*/*w* values) are presented in Table 2. According to the results, the studied strains reached the maximum protein concentration on the 21st day of culture (33–36%, *w*/*w*), except *A. subrufescens* in which protein levels were low (≤10.67%, *w*/*w*). The maximum protein concentration was recorded for *A. bisporus* (36%, *w*/*w*), followed by *P. ostreatus* (34%, *w*/*w*) on the 21st day of culture, but no significant difference was observed between them. However, these values showed a significant difference compared to the maximum protein concentration of the rest two fungi. It is noticeable that *P. ostreatus* on the 6th day of culture had already reached a protein percentage of 8.17% (*w*/*w*), when the rest of the fungi presented values of 0.5–4% (*w*/*w*), being efficient in the production of proteins from the very first stages of culture when biomass concentration was still low. *A. bisporus* exhibited the highest rate of protein production between 6th and 12th day of culture, reaching almost its peak value on 12th day. In regard to protein production of *P. eryngii*, it is noticed that even though the carbon source was not completely consumed on the 21st day of culture, protein content was satisfactory (22.14%, *w*/*w*). Results are comparable to those reported in literature, where according to Chang and Miles [7] the protein content of *P. ostreatus* carposome is between 10.5 and 30.4%, *w*/*w*, while that of mycelial mass ranged from 14.31 to 29.76% *w*/*w* [39]. In addition, the protein content of the mycelia from *P. eryngii* was 24.52% *w*/*w*, which is higher than the protein content (18.91% in pileus and 11.34% *w*/*w* in stipes) of carposomes [66]. In addition, Hwang and Mau [73] reported 37.02% (*w*/*w*) protein content for *A. bisporus* mycelium. These data reveal that the protein content in biomass of *P. ostreatus*, *P. eryngii* and *A. bisporus* produced in the nonoptimized medium of this study was satisfactory. However, Chang et al. [74] reported 15.6% (*w*/*w*) proteins produced in the mycelium of *A. blazei*, a value relatively higher than that of *A. subrufescens*, which was used in the present study. 

In Figure 1, protein biosynthesis versus biomass production is presented for each strain. It is noticed that protein synthesis resembled the profile of biomass production for all fungi except for *A. subrufescens*, where the continuous increase in proteins suggests that the biomass of this strain would have increased further if culture had been continued. For the rest of the fungi, it is observed that towards the end of the culture, while biomass was still increasing, total proteins were decreased, indicating that biosynthesis switched to other structural processes than protein synthesis. It is noteworthy that the *A. bisporus* strain had entered the stationary phase of growth, and this change was accompanied by a decrease in protein levels. For *P. eryngii*, this reduction probably indicates its transition to the stationary phase, even though a substantial amount of glucose was still present.

It is well-established that *Pleurotus* spp. biomass contains a high percentage of protein. With a mean range of protein content from 10.5 to 49% of dry weight, it is important to investigate different ways to increase their production [39]. Furthermore, cultivation of *Agaricus* spp. presents a wide range of protein concentrations depending on medium composition [40,41]. In conclusion, cultivating mushrooms in the mycelial state takes a shorter incubation time, needs less space and has the potential for higher protein content in comparison to carposomes [66,75].

### 3.4. RNA Determination

Total RNA quantities determined by fluorescence spectroscopy are presented in Table 2, while in Figure 2 a representative gel electrophoresis image of crude RNA is provided. 

RNA is located at the forefront bands (lower molecular weight) where it is visually also noticed that its signal increases with biomass increase, reflecting the growth profile of each fungus during batch culture. It has to be mentioned that determined absolute RNA quantities by purification do not reveal actual RNA content, as purification technologies mainly aim to isolation of large RNA molecules for subsequent use in other analytical procedures with loss of short or degraded RNA [76,77]. The objective of the current analysis is mainly the relative changes of RNA along the culture.

Maximum purified RNA quantity per dry weight (mg/g) appeared on the 21st (*A. bisporus* 0.435 and *P. eryngii* 0.379 mg/g) and the 26th day (*A. subrufescens* 0.372 and *P. ostreatus* 0.443 mg/g) of culture (Table 2). It is observed that among fungi, the highest RNA transcription values were recorded for *A. bisporus* and *P. ostreatus*. The RNA percentage of *P. eryngii* the 6th day of culture was almost zero (0.003 mg/g), indicating that production efficiency of this fungus is low at the early stages of culture. It has to be noted that although RNA values per unit biomass (mg/g) tend to be similar at the end of the culture, the actual RNA yield of each fungus varies significantly (Table 2, Figure 2 and Figure 3). 

As is observed in Figure 3, RNA production (mg/g biomass) of *A. bisporus* increased from the 6th to 12th day of culture, entering a steady state on the 21st day. RNA values of *A. subrufescens* increased significantly after 16 days of culture. It is noticed that the reduced metabolic rate throughout the culture period of the certain fungus strain was reflected in the reduced RNA production, which is in accordance with its estimated maximum growth rate being the lowest among the studied fungi. Regarding the strains of genus *Pleurotus*, *P. eryngii*, despite having lower RNA value on day 6 compared to *P. ostreatus*, reached significant levels on the 21st day of culture. *P. ostreatus* showed a steady rise in RNA values throughout the culture, reaching a plateau at the end with the maximum value of all studied strains. For *A. bisporus* and *P. eryngii*, the RNA reduction after the 21st day also indicates the start of the end of the culture.

The RNA profile can serve as an indicator of fungal growth, as RNA synthesis appears earlier, signaling future protein/biomass production. This is supported from the calculated correlation coefficients among RNA vs. biomass and RNA vs. protein, 0.97 for both of them, indicating a very strong relation between RNA and biomass, as well as RNA and the protein-production profile (Figure 4). 

### 3.5. Glucan Measurement

Glucans are carbohydrate ingredients of high interest due to their commercial value. Total glucan production by the four studied fungi during their liquid culture are presented in Table 1. The concentration of glucans varied among the different strains of fungi. Specifically, the maximum values (%, *w*/*w*) were observed in *A. bisporus* after 21 days (8%, *w*/*w*) and *P. ostreatus* after 26 days (6.73%, *w*/*w*) of culture. The max value of *A. bisporus* was significantly different to the corresponding values of the three rest fungi, while the maximum values of *A. subrufescens* and *P. eryngii* were similar (not significantly different), 4.7%, *w*/*w* and were achieved at the end of the cultivation period. Since most of the glucose is consumed during the last stages of fermentation, producing biomass, the positive correlation of glucan production with biomass production was expected (Figure 5). 

A decrease in the concentration of glucans was detected for *A. bisporus* and *P. eryngii*, corresponding to the total IPS decrease at the end of cultivation period. However, *A. subrufescens* was the only fungi that produced a high quantity of glucans by a small amount of biomass. Additionally, the glucan percentage in IPS was calculated and its values ranged from 18 to 33.8% *w*/*w* for *A. bisporus*, 8.3 to 20.4% for *A. subrufescens*, 8.9 to 13.4% *w*/*w* for *P. eryngii* and 13.9 to 31.7% *w*/*w* for *P. ostreatus*. Thus, the IPS of *A. subrufescens*, although present in smaller quantities, consisted of higher percentages of glucans than those of *P. eryngii* (maximum values of 20.4 and 13.4% *w*/*w*, respectively). These data indicate that the mycelium of *A. subrufescens* is quite high in glucans and this fact may increase interest in the wide cultivation of this mushroom [78]. Finally, it has to be mentioned that the assessment of fungi biochemical components as % *w*/*w* mainly reflects metabolic behaviors but may not correspond to final yields. 

In previous studies, the concentration of glucans in the mycelium has been compared to that of the carposomes. According to Nitschke et al. [79], the mycelium of *P. ostreatus* contained 4.64 *w*/*w* of β-glucans, while the carposome contained 9.05; values higher than those of *A. bisporus* that were 3.79 and 6.00 *w*/*w*, respectively. Toledo et al. [80] analyzed the carposome of various mushrooms, including those examined in the present study. In particular, their growth was achieved on a solid substrate and an enzymatic method was used to extract the glucans. These authors reported that *P. ostreatus* contained larger amounts of glucans, followed by *P. eryngii* and *A. subrufescens*, which had similar values, and then *A. bisporus.* Finally, the mycelium of *P. eryngii* was the poorest in glucans, compared to those of the above fungi (3.34, *w*/*w*). Total glucan-production patterns of the four fungi in relation to biomass are presented in Figure 5. It is noticed that glucan production coincides with biomass production throughout the culture in all cases. In addition, as previously stated, production of glucans seems to be species- and not genus-dependent. *A. bisporus* appeared to be the most efficient of the studied strains in regard to its content of glucans. On the contrary, *P. ostreatus*, having consumed 99% of the carbon substrate at the end of the culture, produced the maximum biomass and had the highest yield of glucans among all strains. Production of polysaccharides such as glucans from *P. ostreatus* species is important and is under thorough investigation not only for their nutritional properties, but also for their biological value in medicinal applications. They are investigated as immunomodulators, antigens in the formulation of vaccines for cancer therapies, as well as for cholesterol reduction [12,13,14]. It is worth mentioning that *A. subrufescens* on the 26th day of culture exhibited a higher percentage of glucans than *P. eryngii*, despite the fact that its biomass production was almost half that of *P. eryngii.* These data indicate that the mycelium of *A. subrufescens*, contrary to its low biomass production, is efficient in glucan synthesis, and this is one of the reasons for the increasing interest in the cultivation of this mushroom [78]. β-glucans can operate as prebiotics and dietary fibers and reduce pathogen expansion by enhancing the growth of probiotic bacteria in the intestine, improving human health status [72].

### 3.6. Glucanase Activity Measurement

Glucanases are enzymes that are specialized in the hydrolysis of specific glycosidic bonds. Endo-1,3-β-D glucanase production of the four fungi, expressed as units of enzyme activity per g biomass, is presented in Table 2. All fungi showed increasing glucanase activity up to the 21st day of culture, but *A. bisporus* and *P. ostreatus* (12.74 and 11.67 Units/g biomass, respectively, not significantly different) had reached the certain day with almost two-fold higher activity, which is significantly different in comparison to the rest of the fungi. Their maximum glucanase activity coincides with their maximum protein concentration and RNA quantity of that day. *A. subrufescens* had much lower and delayed production of glucanases until the 16th day. Instead, *P. eryngii* showed a low increase in the enzyme activity values presenting a maximum at 26th day. It appears that the production of glucanases is a process related to the growth stage of the fungus, as these enzymes were highly produced during the later log and the stationary phase of biomass production. The activity of the fungal glucanases might be triggered in response to carbon-source reduction in the growth medium. These enzymes hydrolyze glycosidic linkages of sugars, thereby might release glucose units in their environment from fungal cell walls. In the present study, glucose was used as a main carbon source, consumed during culture to produce biomass. It seems that fungi perceive the decrease in available glucose levels and scavenge carbon sources by increased enzymatic activity. 

As has been already mentioned, the production of glucanases increased from the first days of cultivation when the carbon source was not a limiting factor (Figure 6). This phenomenon is observed for all fungi up to the 21st day. In the case of *A. bisporus* and *P. ostreatus*, the total glucose amount in the growth media was mostly consumed on the last day of culture, while *A. subrufescens* and *P. eryngii* had consumed around half of their glucose source at the same day.

### 3.7. Extrapolated Optimization

It is noticed from the data of the batch fungal cultures that there is significant variability in the production of glucans and glucanases among the studied fungi. Thus, optimization for future continuous cultures will need large-scale experimental designs that become cumbersome and costly. As a first step for optimizing future continuous production, batch-culture results were further statistically analyzed using response 3D surfaces and contour graphics to extrapolate potential culture parameters [81]. Results of this analysis are presented in Figure 7, where glucans and glucanases are plotted against glucose consumption and the corresponding biomass-production rate in order to determine optimal dilution and flow rates in future continuous cultures. Contours near the red areas in each studied case reflect potential increased production of glucans and glucanases in permissible conditions, indicating possible glucose concentrations and the corresponding biomass-production rate that needs to be attained. The most promising strain for the parallel production of glucans and glucanase according to contour-graphics analysis tends to be *A. bisporus*, exhibiting a wider flexibility in the combinations of biomass-production rate with carbon-source-utilization rate, while *P. ostreatus* shows the potency for the highest glucanase production of all strains. However, these observations need further experimental verification in continuous cultures.

## 4. Conclusions

All fungi showed the ability to produce glucans and glucanases. Based on the results from the batch cultures and extrapolated-data analysis, the *A. bisporus* strain seems to be the most efficient in glucan and glucanase (endo-1,3-β-D-glucanase) production. RNA analysis also showed that RNA production can be a valuable prognostic biomarker of successful cultures, as it is strongly correlated with fungal growth. The yields of the fungi in terms of the studied characteristics are not related to the genera of the fungi, but each fungus is an independent case. Since the medium and culture conditions were identical in all cases, differences observed in the biomass production, growth rate and substrate consumption reflect mainly the genetic diversity of the fungal strains. However, this conclusion would be clear if more species of fungi were compared per genus.

## Figures and Tables

**Figure 1 microorganisms-10-00964-f001:**
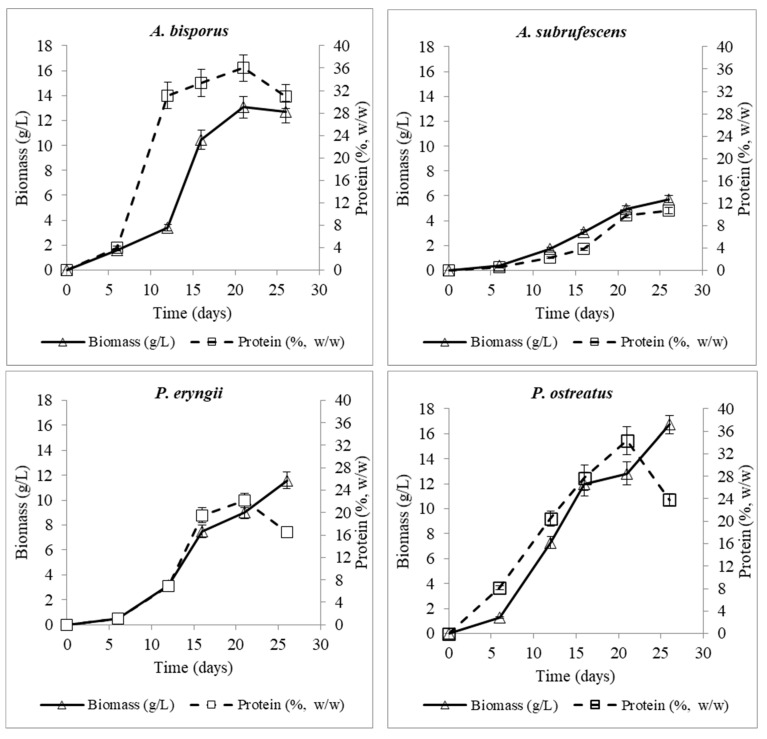
Production of biomass and total proteins of *A. bisporus*, *A. subrufescens*, *P. eryngii* and *P. ostreatus* during their culture in agitated (120 ± 5 rpm) flasks of C/N molar ratio 20, at 26 ± 1 °C.

**Figure 2 microorganisms-10-00964-f002:**
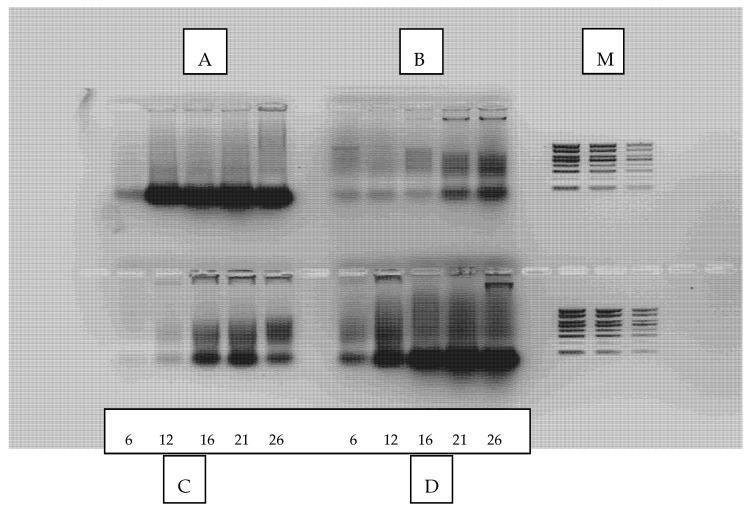
Representative electrophoresis in 2% agarose gel of crude RNA released from lysed fungal cells of (A) *A. bisporus*, (B) *A. subrufescens*, (C) *P. eryngii*, (D) *P. ostreatus.* (M) refers to DNA marker (1kb).

**Figure 3 microorganisms-10-00964-f003:**
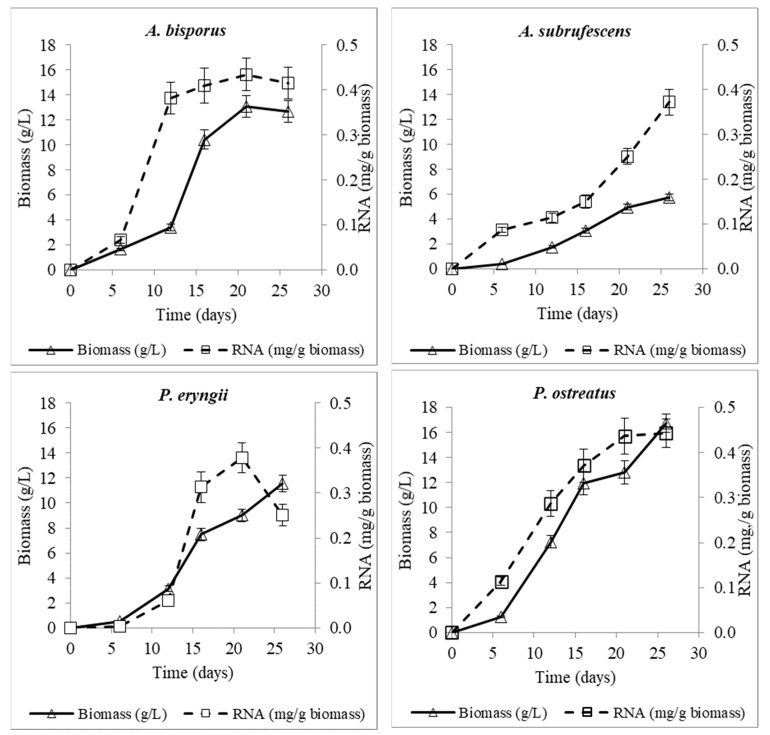
Produced biomass and RNA levels of *A. bisporus*, *A. subrufescens*, *P. eryngii* and *P. ostreatus* during their cultivation in agitated (120 ± 5 rpm) liquid cultures of C/N molar ratio 20, at 26 ± 1 °C.

**Figure 4 microorganisms-10-00964-f004:**
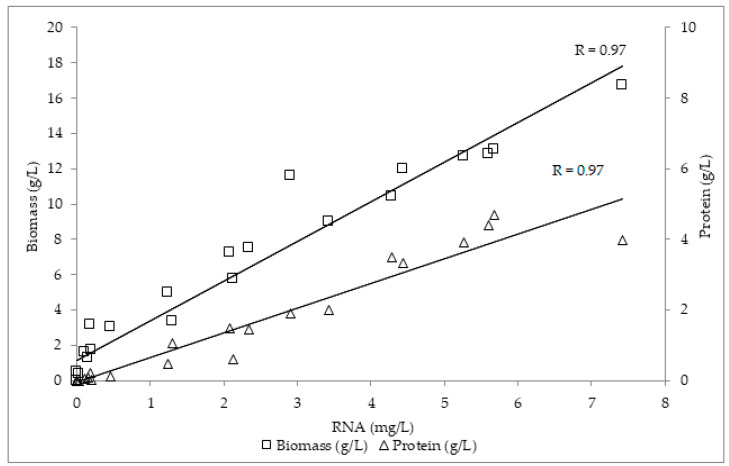
Correlation among RNA vs. biomass and RNA vs. protein of *A. bisporus*, *A. subrufescens*, *P. eryngii* and *P. ostreatus.*

**Figure 5 microorganisms-10-00964-f005:**
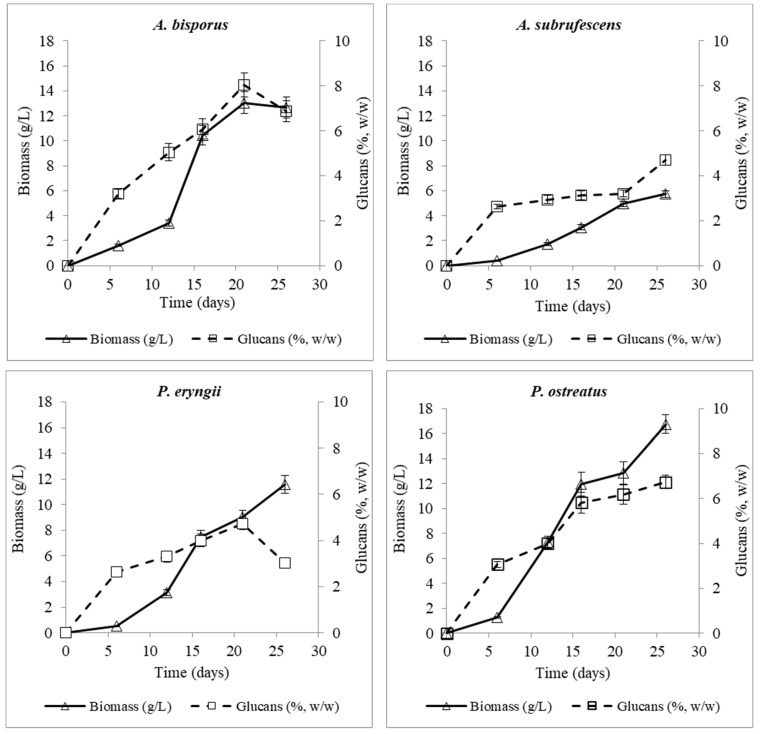
Total glucan-production patterns of *A. bisporus*, *A. subrufescens*, *P. eryngii* and *P. ostreatus* in relation to their produced biomass.

**Figure 6 microorganisms-10-00964-f006:**
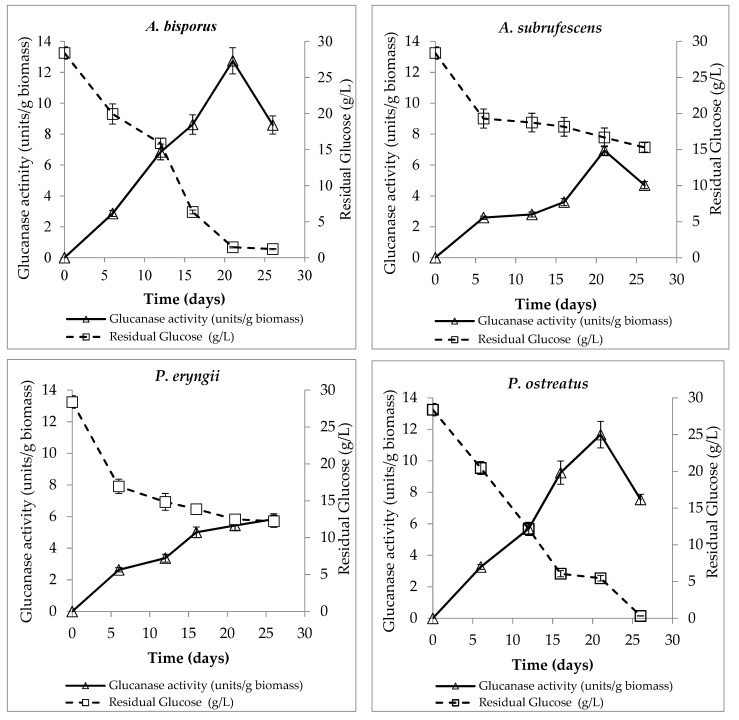
Residual glucose and glucanase activity of *A. bisporus*, *A. subrufescens*, *P. eryngii* and *P. ostreatus* during their cultivation in agitated (120 ± 5 rpm) liquid cultures of C/N molar ratio 20, at 26 ± 1 °C.

**Figure 7 microorganisms-10-00964-f007:**
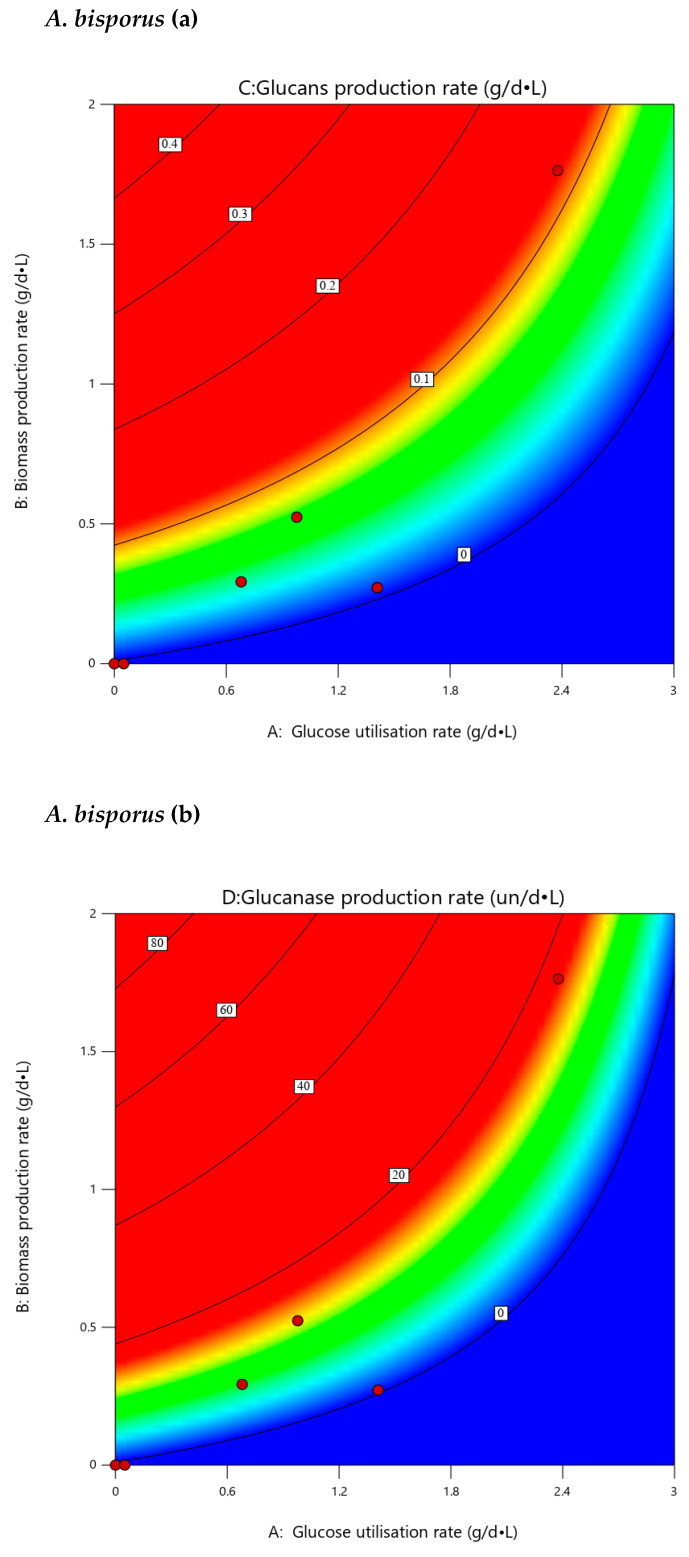
Contour analysis of glucans (a) and endo-1,3-β-D-glucanases (b) production of *A. bisporus*, *A. subrufescens*, *P. eryngii* and *P. ostreatus* against Glucose utilization rate (A) and the corresponding biomass-production rate (B; g: grams, d: days, L: Liter). Contour areas present glucans and glucanase production levels for different combinations of culture parameters that can determine flow and dilution rates in future continuous cultures.

**Table 1 microorganisms-10-00964-t001:** Biochemical characteristics during the cultivation of *A. bisporus* (a), *A. subrufescens* (b), *P. eryngii* (c) and *P. ostreatus* (d) in agitated (120 ± 5 rpm) liquid cultures of C/N molar ratio 20, at 26 ± 1 °C.

Fungal Species	Time (Days)	Biomass (g/L)	Residual Glucose (g/L)	Yx/Gl_c_ (g/g)	IPS (g/L)	Glucans (g/L)	IPS (%, *w*/*w*)	Glucans (%, *w*/*w*)
*A. bisporus*	0	0.00 ± 0.00 *	28.39 ± 0.85	-	0.00 ± 0.00	0.00 ± 0.00	0.00 ± 0.00	0.00 ± 0.00
	6	1.63 ± 0.11	19.93 ± 1.39	0.19 ± 0.04	0.29 ± 0.02	0.05 ± 0.00	17.78 ± 1.15	3.20 ± 0.21
	12	3.39 ± 0.25	15.85 ± 0.71	0.27 ± 0.02	0.74 ± 0.06	0.17 ± 0.01	21.91 ± 1.63	5.05 ± 0.38
	16	10.44 ± 0.77	6.34 ± 0.27	0.47 ± 0.03	3.43 ± 0.25	0.63 ± 0.05	32.82 ± 2.41	6.08 ± 0.45
	21	13.06 ± 0.87 ^b,d^	1.45 ± 0.09	0.48 ± 0.04	3.21 ± 0.21	1.05 ± 0.07	24.57 ± 1.64	8.05 ± 0.54 ^b,c,d^
	26	12.67 ± 0.86	1.20 ± 0.08	0.47 ± 0.02	2.57 ± 0.18	0.87 ± 0.06	20.32 ± 1.38	6.87 ± 0.47
*A. subrufescens*	0	0.00 ± 0.00	28.39 ± 0.85	-	0.00 ± 0.00	0.00 ± 0.00	0.00 ± 0.00	0.00 ± 0.00
	6	0.40 ± 0.02	19.30 ± 1.33	0.04 ± 0.03	0.08 ± 0.00	0.01 ± 0.00	19.50 ± 0.79	2.62 ± 0.11
	12	1.73 ± 0.10	18.74 ± 1.29	0.18 ± 0.02	0.61 ± 0.04	0.05 ± 0.00	35.29 ± 2.12	2.94 ± 0.18
	16	3.06 ± 0.20	18.16 ± 1.31	0.30 ± 0.03	0.70 ± 0.04	0.10 ± 0.01	22.92 ± 1.47	3.12 ± 0.20
	21	4.96 ± 0.21	16.68 ± 1.31	0.42 ± 0.04	1.39 ± 0.06	0.16 ± 0.01	28.07 ± 1.20	3.21 ± 0.14
	26	5.73 ± 0.28 ^a,c,d^	15.30 ± 0.67	0.44 ± 0.04	1.32 ± 0.07	0.27 ± 0.01	25.01 ± 1.24	4.70 ± 0.23 ^a,d^
*P. eryngii*	0	0.00 ± 0.00	28.39 ± 0.85	-	0.00 ± 0.00	0.00 ± 0.00	0.00 ± 0.00	0.00 ± 0.00
	6	0.51 ± 0.03	16.96 ± 0.97	0.04 ± 0.01	0.12 ± 0.01	0.01 ± 0.00	23.53 ± 1.22	2.63 ± 0.14
	12	3.15 ± 0.23	14.85 ± 1.14	0.23 ± 0.02	1.17 ± 0.09	0.10 ± 0.01	37.25 ± 2.73	3.31 ± 0.24
	16	7.49 ± 0.51	13.88 ± 0.60	0.52 ± 0.03	2.84 ± 0.19	0.30 ± 0.02	37.97 ± 2.58	4.00 ± 0.27
	21	9.03 ± 0.50	12.50 ± 0.53	0.57 ± 0.04	3.20 ± 0.18	0.43 ± 0.02	35.42 ± 1.97	4.74 ± 0.26 ^a,d^
	26	11.58 ± 0.67 ^b,d^	12.23 ± 0.84	0.72 ± 0.05	3.82 ± 0.22	0.35 ± 0.02	32.96 ± 1.91	3.03 ± 0.18
*P. ostreatus*	0	0.00 ± 0.00	28.39 ± 0.85	-	0.00 ± 0.00	0.00 ± 0.00	0.00 ± 0.00	0.00 ± 0.00
	6	1.27 ± 0.06	20.46 ± 0.88	0.16 ± 0.01	0.37 ± 0.02	0.04 ± 0.00	22.04 ± 1.01	3.06 ± 0.14
	12	7.26 ± 0.48	12.16 ± 0.91	0.45 ± 0.02	1.71 ± 0.11	0.29 ± 0.02	23.61 ± 1.56	4.00 ± 0.27
	16	11.96 ± 0.96	6.06 ± 0.42	0.54 ± 0.02	3.32 ± 0.27	0.69 ± 0.06	27.73 ± 2.22	5.81 ± 0.46
	21	12.82 ± 0.93	5.45 ± 0.39	0.56 ± 0.03	2.62 ± 0.19	0.79 ± 0.06	20.41 ± 1.48	6.19 ± 0.45
	26	16.75 ± 0.73 ^a,b,c^	0.33 ± 0.01	0.60 ± 0.04	3.56 ± 0.15	1.13 ± 0.05	21.23 ± 0.93	6.73 ± 0.29 ^a,b,c^

* Values are the averages of three replicates with standard deviation <10%. ^a, b, c, d^ correspond to used fungal strains. Presence of letters a–d denotes significant differences (*t*-test, *p* < 0.05) in comparisons of biochemical compounds between different strains.

**Table 2 microorganisms-10-00964-t002:** Production of RNA, protein and glucanase during cultivation of *A. bisporus* (a), *A. subrufescens* (b), *P. eryngii* (c) and *P. ostreatus* (d) in agitated (120 ± 5 rpm) liquid cultures of C/N molar ratio 20, at 26 ± 1 °C.

Fungal Species	Time (Days)	RNA (mg/L)	Protein (g/L)	Glucanase Activity (Units/L)	RNA (mg/g Biomass)	Protein (%, *w*/*w*)	Glucanase Activity (Units/g Biomass)
*A. bisporus*	0	0.000 ± 0.000 *	0.000 ± 0.000	0.000 ± 0.000	0.000 ± 0.000	0.000 ± 0.000	0.000 ± 0.000
	6	0.109 ± 0.012	0.065 ± 0.004	4.668 ± 0.929	0.067 ± 0.007	4.000 ± 0.260	2.862 ± 0.186
	12	1.298 ± 0.119	1.055 ± 0.078	23.198 ± 2.542	0.383 ± 0.035	31.119 ± 2.312	6.843 ± 0.508
	16	4.280 ± 0.403	3.487 ± 0.256	89.959 ± 4.158	0.410 ± 0.039	33.400 ± 2.448	8.617 ± 0.632
	21	5.681 ± 0.473	4.702 ± 0.314	166.439 ± 4.250	0.435 ± 0.036	36.006 ± 2.402 ^b,c^	12.744 ± 0.850 ^b,c^
	26	5.258 ± 0.446	3.921 ± 0.267	108.826 ± 4.295	0.415 ± 0.035	30.947 ± 2.107	8.589 ± 0.585
*A. subrufescens*	0	0.000 ± 0.000	0.000 ± 0.000	0.000 ± 0.000	0.000 ± 0.000	0.000 ± 0.000	0.000 ± 0.000
	6	0.035 ± 0.002	0.002 ± 0.000	1.037 ± 0.105	0.087 ± 0.006	0.522 ± 0.021	2.593 ± 0.105
	12	0.197 ± 0.019	0.041 ± 0.002	4.847 ± 0.824	0.114 ± 0.011	2.342 ± 0.141	2.800 ± 0.168
	16	0.459 ± 0.047	0.116 ± 0.007	11.023 ± 1.092	0.150 ± 0.015	3.800 ± 0.244	3.600 ± 0.231
	21	1.245 ± 0.085	0.488 ± 0.021	34.425 ± 1.481	0.251 ± 0.017	9.828 ± 0.420	6.938 ± 0.296 ^a,c,d^
	26	2.133 ± 0.169	0.612 ± 0.030	26.944 ± 1.166	0.372 ± 0.030	10.675 ± 0.529 ^a,c,d^	4.700 ± 0.233
*P. eryngii*	0	0.000 ± 0.000	0.000 ± 0.000	0.000 ± 0.000	0.000 ± 0.000	0.000 ± 0.000	0.000 ± 0.000
	6	0.002 ± 0.000	0.006 ± 0.000	1.343 ± 0.108	0.003 ± 0.000	1.117 ± 0.058	2.633 ± 0.136
	12	0.189 ± 0.023	0.218 ± 0.016	10.629 ± 1.236	0.060 ± 0.007	6.909 ± 0.506	3.372 ± 0.247
	16	2.344 ± 0.253	1.463 ± 0.099	37.489 ± 1.699	0.313 ± 0.034	19.535 ± 1.326	5.005 ± 0.340
	21	3.422 ± 0.301	2.000 ± 0.111	49.111 ± 1.509	0.379 ± 0.033	22.145 ± 1.229 ^a,b,d^	5.439 ± 0.302
	26	2.907 ± 0.270	1.911 ± 0.111	67.646 ± 1.694	0.251 ± 0.023	16.501 ± 0.957	5.842 ± 0.339 ^a,b,d^
*P. ostreatus*	0	0.000 ± 0.000	0.000 ± 0.000	0.000 ± 0.000	0.000 ± 0.000	0.000 ± 0.000	0.000 ± 0.000
	6	0.145 ± 0.012	0.104 ± 0.005	4.141 ± 0.748	0.114 ± 0.009	8.168 ± 0.375	3.259 ± 0.150
	12	2.085 ± 0.211	1.487 ± 0.098	41.352 ± 1.876	0.287 ± 0.029	20.462 ± 1.348	5.692 ± 0.375
	16	4.437 ± 0.444	3.318 ± 0.265	110.651 ± 3.696	0.371 ± 0.037	27.742 ± 2.217	9.252 ± 0.739
	21	5.602 ± 0.506	4.409 ± 0.319	149.565 ± 4.217	0.437 ± 0.039	34.392 ± 2.487 ^b,d^	11.667 ± 0.843 ^b,c^
	26	7.420 ± 0.518	3.991 ± 0.174	126.433 ± 1.646	0.443 ± 0.031	23.827 ± 1.039	7.548 ± 0.329

* Values are the averages of three replicates with standard deviation <10%.^a, b, c, d^ correspond to used fungal strains. Presence of letters a–d denotes significant differences (*t*-test, *p* < 0.05) in comparisons of biochemical compounds between different strains.

## Data Availability

The detailed results of the study could be provided by D.A. or P.D. after request.

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
