# Peer review of "Biochemical Evaluation of Agaricus and Pleurotus Strains in Batch Cultures for Production Optimization of Valuable Metabolites"

_microorganisms, 2022, doi:10.3390/microorganisms10050964_

Round 1
Reviewer 1 Report
After the review process, I have several comments: - the authors should insert references in all Materials and Methods sections; - If the authors used " significantly", they should insert the value of the "p"; - figure 7 should be included as supplementary material; - the discussions are poor in correlation with previous papers that link the polysaccharides obtained by batch cultivation of Pleurotus ostreatus mycelium with other biological properties My general recommendation is to accept this paper after these modifications.
Author Response
Please, see attached file.

Reviewer 2 Report
Thank you for your interesting research. Some points must be carefully revised:
- Line 42. “Low-fat content”. Could you indicate a range of fat content for Agaricus genus?
- Lines 52-53. Perhaps range of contents should be included for the mentioned compounds (to enrich this section).
- Since mushroom polysaccharides (particularly β-glucans) are relevant compounds for this research work, some of their interesting biological activities must be mentioned through the INTRODUCTION SECTION. For instance, hypocholesterolemic activity reported for several mushroom β-glucans. Useful references: 10.1615/IntJMedMushrooms.2017024413 ; https://doi.org/10.1039/C9FO01744E
- M&M: Section 2.1. Why 26 days?
- M&M: Section 2.7. Determination of total β-glucans. If the commercial Megazyme was used, it would be appropriate to indicate that total glucans and total α-glucans are measured and total β-glucans are calculated by substraction.
- RESULTS: Lines 228-229 / Table 1. Would biomass production be even higher later than 26th day for ostreatus?
- RESULTS: Line 219: You stated that β-glucans were calculated. However, Table 1 showed (total?) “glucans” values and section 3.5 and Figure 5 are also referred to glucans (instead of β-glucans). Could you please revise it for clarification?
- The quantitative results that were obtained in this work must be subjected to statistical analyses to confirm if the differences between the values are significant. The utilized statistics method/software must be included in M&M section.
Reviewer 3 Report
I have carefully and with pleasure read your manuscript entitled “Biochemical Evaluation of Agaricus and Pleurotus Strains in Batch Cultures for Production Optimization of Valuable Metabolites”. In the literature, there is a continuous development of research into the possibilities of fungi to produce compounds beneficial for our health. Each such research is precious. Below are my suggestions:
- please add in the abstract that the measurements were made in 6 periods (0, 6, 12, 16, 21 and 26 days); this is a great value for the manuscript;
- please, add the definition of Yx/Glc in the table description;
- in my version the figure 7 quality is poor, maybe it is possible to combine the pictures to one picture ale placed it only on one page?
Best regards,
Reviewer
Reviewer 4 Report
The article is well written and discusses an interesting topic in the field of fermentation. A lot of data is presented (maybe a little too much) but unfortunately it is not interpreted from a statistical point of view. It is not enough to present tables / graphs only with values and Standard dev .. In my opinion, it would be useful for some data to be compared / interpreted statistically (some results / most relevant from different fermentation days) and to discuss the results obtained from statistical interpretations in the text.
Round 2
Reviewer 1 Report
Dear Authors,
In this phase, it will be good for data interpretation to add more comments about the therapeutic properties of edible mushrooms in gut microbiota modulation.
Best regards!
